

# A Numerical Study of Near Inertial Motions in Mid-Atlantic Bight Area Induced by Hurricane Irene (2011)

Peida Han[1] and Xiping Yu[2]

## Abstract

Hurricane Irene generated strong near inertial currents (NICs) in the ocean waters when passing over the Mid-Atlantic Bight (MAB) of the U. S. East Coast in late August 2011. It is demonstrated that a combination of the valuable field data with detailed model results can be exploited to study the development and decay mechanism of this event. Numerical results obtained with regional oceanic modeling system (ROMS) are shown to agree well with the field data. Both computed and observed results show that the NICs were significant in most areas of the MAB region except in the nearshore area where the stratification was totally destroyed by the hurricane-induced strong mixing. Based on the energy budget, it is clarified that the near inertial kinetic energy (NIKE) was mainly gained from the wind power during the hurricane event. In the deep water region, NIKE was basically balanced by the vertical turbulence diffusion (40%) and downward divergence (33%). While in the continental shelf region, NIKE was mainly dissipated by the vertical turbulence diffusion (67%) and partially by the bottom friction (24%). Local dissipation of NIKE due to turbulence diffusion is much more closely related to the rate of the vertical shear rather than the intensity of turbulence. The strong vertical shear at the offshore side of the continental shelf leaded to a rapid dissipation of NIKE in this region.

**Keywords:** Hurricane Irene; Mid-Atlantic Bight; Near inertial current; Energy budget; Timescale of near inertial energy decay

---

[1] PhD Candidate. Department of Hydraulic Engineering, Tsinghua University, Beijing, China.

[2] Corresponding Author. Professor. Department of Ocean Science and Engineering, Southern University of Science and Technology, Shenzhen, China. Email: yuxp@sustech.edu.cn



## 1. Introduction

Near inertial currents (NICs), observed widely in ocean basins around the world, are characterized by the important role of Coriolis effect and by the periodic motion with the frequency of an inertial mode (Garrett, 2001). The basic energy source of these freely flowing currents is the wind power (Pollard, 1980; D'Asaro et al., 1985). Globally, the annually averaged wind power supply to NICs was estimated ranging from 0.3 TW to more than 1 TW by previous investigators (Alford, 2003a; Furuichi et al., 2008; Rimac et al., 2013). As a comparison, the total power required to maintain the abyssal stratification and the thermohaline circulation is about 2 TW (Munk and Wunsch, 1998). This implies that NIC is a very important phenomenon in physical oceanography. In fact, NICs are believed to have a significant role in upper-ocean mixing, which may substantially affect the thermohaline circulation and even modulate the climate (Gregg, 1987; Alford, 2003b; Jochum et al., 2013).

A tropical or an extratropical cyclone (hereinafter collectively referred as TC) is a rotating low-pressure and strong-wind mesoscale weather system, which generates NICs more powerfully than other types of atmospheric processes in nature (Alford et al., 2016; Steiner et al., 2017). When a TC passes over a deep ocean, enormous energy is directly transferred into the ocean waters, which rapidly generates strong NICs with a velocity up to 1 m/s in the horizontal direction of the mixed layer (Price, 1983; Sanford et al., 2011). Right-bias effect is often shown in the NIC pattern, i.e., NICs are more intense on the right side of the hurricane track, due to the resonance between the surface flow driven by NICs and clockwise rotating wind stress on the right side (Chang and Anthes, 1978; Price, 1994). After the passage of a TC, the surface near inertial energy usually persists for several inertial cycles, and then gradually decays (Price, 1983; Sanford et al., 2011; Hormann et al., 2014; Zhang et al., 2016; Wu et al., 2020).

It is known that NICs in shallow waters show some significant differences with those in deep waters and the velocity of NICs in shallow waters is usually of a smaller magnitude of 0.1-0.5 m/s (Chen and Xie, 1997; Rayson et al., 2015; Yang et al., 2015; Chen et al., 2017;



Zhang et al., 2018). The decrease of current velocity in shallow waters may be an effect of
the sea-bottom friction as Rayson et al. (2015) pointed out. Chen and Xie (1997), however,
found that it was because a significant part of the wind input, which may otherwise be an
energy source of the NICs, was exhausted to generate a wave-induced nearshore current
system. Chen et al. (2017) considered that barotropic waves in the shallow waters, such as
seiches, may trap some wind energy. In addition to the difference in magnitude, the modes
of the NICs in shallow and deep waters are also different. More specifically, a two-layer
structure was observed in shallow waters in several studies, i.e., NICs were in opposite
phases in surface and bottom layers, which differed from the conventional multi-layer mode
in deep waters (Chen et al., 1996; Shearman, 2005; Yang et al., 2015), though a multi-layer
mode may also be observed sometimes in nearshore waters due to combined effect of
changing wind stress, variable stratification and nonlinear bottom friction (Mackinnon and
Gregg, 2005).

There have been a considerable number of studies on the decay of specific TC

generated NICs in coastal regions. Rayson et al. (2015) paid attention to four intense TCs
on the Australian North-West Shelf and related the rapid decay of NICs in shallow waters
to the bottom friction. Yang et al. (2015) examined coastal ocean responses to Typhoon
Washi and found that the negative background vorticity could trap near inertial energy and
result in a slow decay. Shen et al. (2017) investigated five TCs over the Taiwan Strait and
identified a rapid decaying rate due to nonlinear interaction between NICs and tides. Zhang
et al. (2018) studied Hurricane Arthur in Mid-Atlantic Bight and showed that excessive wind
input does not necessarily lead to amplification of NICs because intensive wind input is
usually accompanied by an even higher rate of energy dissipation.

Though a significant number of investigations have been conducted, some basic

features of a TC induced NIC in the coastal ocean are still not clarified. For instance, the
energy budget in the NIC generated by a TC has not yet been thoroughly discussed in either
deep or shallow waters; and the relative importance of different physical processes including
advection, conversion, turbulence diffusion, bottom friction, energy divergence, etc., in the





energy budget has not yet been fully understood. In addition, it is still not concluded on
which processes dominate the decay of near inertial energy or on how each physical process
affects the decay rate of the near inertial energy in deep and shallow waters, respectively.
Our limited understanding to the basic features of a TC induced NIC is largely due to the
difficulties in ocean observations under extreme weather.

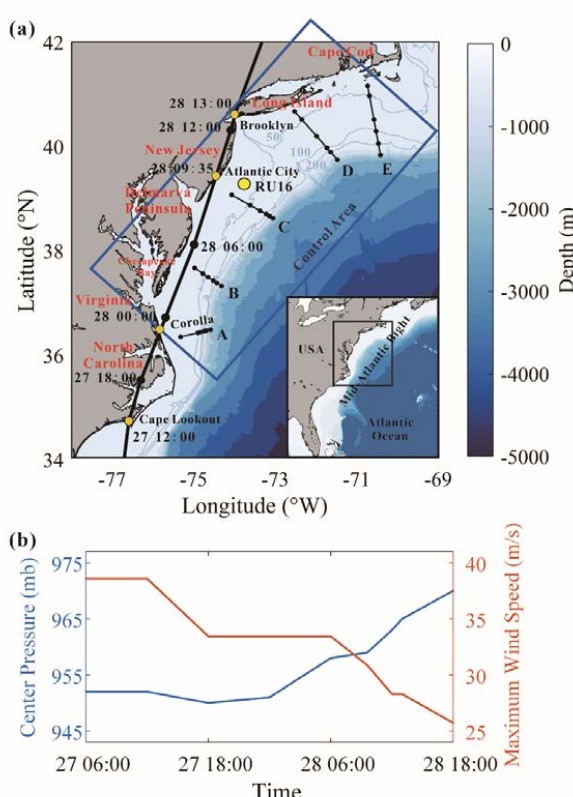


Figure 1. (a) Map of the MAB region. Best track of Hurricane Irene (2011) reported by Avila
and Cangialosi (2011) is shown by a black line. Reanalysis data provided by H*WIND
shows a similar track with Avila and Cangialosi (2011) and is thus omitted. The mean
position of Glider RU16 is marked by a yellow circle. Five virtual sections defined in
Section 3 are marked by short black lines. The control domain defined in Section 4 is marked
by a blue box. (b) Time series of center pressure and 10-m maximum wind speed of
Hurricane Irene reported by Avila and Cangialosi (2011).





In this study, we pay a close attention to the NIC induced by Hurricane Irene (2011).
Hurricane Irene (2011) crossed over the Mid-Atlantic Bight (MAB), a coastal region of the
North Atlantic, extending from Cape Cod, Massachusetts to Cape Lookout, North Caroline,
USA, as shown in Figure 1a. Before the hurricane event, seawater stratification in MAB
was quite strong due to the Cold Pool effect (Lentz, 2017) and the temperature difference
between the surface and the bottom exceeded 10 °C. During the passage of Hurricane Irene
(2011), a network of High-frequency (HF) radars measured the surface currents in MAB
(Roarty et al., 2010). Meanwhile, a Slocum glider launched near New Jersey measured the
vertical profiles of the temperature and the salinity (Schofield et al., 2010). Combination of
the valuable field data with effective numerical techniques then provided an opportunity to
achieve a comprehensive study of the NICs generated by this hurricane event.

## 2. Numerical Model

2.1 Basic Equations
In this study, the ocean responses to Hurricane Irene (2011) are studied using the
regional oceanic modeling system (ROMS) (Shchepetkin and McWilliams, 2005;
Haidvogel et al., 2008). ROMS deals with the Reynolds-averaged N-S equations in the $\sigma$
coordinate system (Freeman et al., 1972). Specifically, the Cartesian coordinate $z$ is
replaced by $\sigma$ based on a general relation $\chi(\sigma) = (z - \eta)/D$, where $\eta$ is the vertical
displacement of the free surface and $D$ is the instantaneous water depth, while $\chi(\sigma)$ is
a stretching function introduced for grid refinement. In the $\sigma$-coordinate system the
Reynolds-averaged N-S equations may finally be expressed as
$$\frac{\partial \xi}{\partial t} + \frac{\partial(\xi u)}{\partial x} + \frac{\partial(\xi v)}{\partial y} + \frac{\partial(\xi \omega)}{\partial \sigma} = 0 \tag{1}$$

$$\begin{aligned} &\frac{\partial(\xi u)}{\partial t} + \frac{\partial(\xi uu)}{\partial x} + \frac{\partial(\xi uv)}{\partial y} + \frac{\partial(\xi u\omega)}{\partial \sigma} - f\xi v + \frac{\xi}{\rho}\frac{\partial p}{\partial x} \\ &= -g\xi\left(\chi\frac{\partial D}{\partial x} + \frac{\partial \eta}{\partial x}\right) + \frac{\partial}{\partial \sigma}\left(\frac{\nu}{\xi}\frac{\partial u}{\partial \sigma}\right) + \frac{\partial}{\partial x}\left(\xi\nu'\frac{\partial u}{\partial x}\right) + \frac{\partial}{\partial y}\left(\xi\nu'\frac{\partial u}{\partial y}\right) \end{aligned} \tag{2}$$



$$\frac{\partial\left(\xi v\right)}{\partial t}+\frac{\partial\left(\xi uv\right)}{\partial x}+\frac{\partial\left(\xi vv\right)}{\partial y}+\frac{\partial\left(\xi v\omega\right)}{\partial\sigma}+f\xi u+\frac{\xi}{\rho}\frac{\partial p}{\partial y}$$
$$=-g\xi\left(\chi\frac{\partial D}{\partial y}+\frac{\partial\eta}{\partial y}\right)+\frac{\partial}{\partial\sigma}\left(\frac{\nu}{\xi}\frac{\partial v}{\partial\sigma}\right)+\frac{\partial}{\partial x}\left(\xi\nu'\frac{\partial v}{\partial x}\right)+\frac{\partial}{\partial y}\left(\xi\nu'\frac{\partial v}{\partial y}\right)$$
(3)

$$0=-\frac{1}{\rho}\frac{\partial p}{\partial\sigma}-g\xi$$
(4)

$$\frac{\partial\left(\xi C\right)}{\partial t}+\frac{\partial\left(\xi uC\right)}{\partial x}+\frac{\partial\left(\xi vC\right)}{\partial y}+\frac{\partial\left(\xi\omega C\right)}{\partial\sigma}$$
$$=\frac{\partial}{\partial\sigma}\left(\frac{\kappa}{\xi}\frac{\partial C}{\partial\sigma}\right)+\frac{\partial}{\partial x}\left(\xi\kappa'\frac{\partial C}{\partial x}\right)+\frac{\partial}{\partial y}\left(\xi\kappa'\frac{\partial C}{\partial y}\right)$$
(5)

where, $\xi=\partial z/\partial\sigma=D\left(\partial\chi/\partial\sigma\right)$; $u$, $v$, $\omega$ are the velocity components in $x$, $y$, $\sigma$
directions, respectively; $C$ stands for the potential temperature $T$ or salinity $S$; $p$ is
the seawater pressure; $\rho$ is the density of the seawater; $f=2\Omega\sin\phi$ is the Coriolis
parameter with $2\Omega=1.458\times10^{-4}\ \text{s}^{-1}$ and $\phi$ being the latitude; $\nu$ and $\kappa$ are the
diffusion coefficients for momentum and potential temperature or salinity, respectively, in
the vertical direction; $\nu'$ and $\kappa'$ are those in the horizontal directions; Note that Eq. (1)
is the continuity equation; Eqs. (2) and (3) are equations of motion in two horizontal
directions; Eq. (4) is the hydrostatic assumption; Eq. (5) is the advection-diffusion equation
of the potential temperature or the salinity. The density of the seawater $\rho$ is determined
following the equation of state proposed by Jackett and McDougall (1995):
$$\rho\left(S,T,p\right)=\frac{\rho_0}{1-p/K\left(S,T,p\right)}$$
(6)

where $\rho_0=\rho\left(S,T,0\right)$ is the seawater density at the standard atmospheric pressure and
$K\left(S,T,p\right)$ is the bulk modulus, both are given by Jackett and McDougall (1995).
The vertical mixing is known to play an important role in determining the structure of
a NIC, so it must be properly evaluated. In this study, we consider $\nu=\nu_0+\nu_e$ and
$\kappa=\kappa_0+\kappa_e$, in which $\nu_0$ and $\kappa_0$ are the molecular viscosity and diffusivity of the
seawater, set to $\nu_0=10^{-5}\ \text{m}^2/\text{s}$ and $\kappa_0=10^{-6}\ \text{m}^2/\text{s}$ following previous suggestions (Xu
et al., 2002; Li and Zhong, 2007; Lentz, 2017), while $\nu_e$ and $\kappa_e$ are the eddy viscosity
and diffusivity, determined by the conventional k-ε turbulence model (see Rodi (1987) and





Umlauf and Burchard (2003) for detailed description), a widely employed model that
demonstrated good performance in simulating various oceanographic processes
(Olabarrieta et al., 2011; Toffoli et al., 2012; Zhang et al., 2018).

Horizontal mixing is included in Eqs. (2), (3) and (5), though it has been pointed out

to play a relatively insignificant role in simulating response of the stratified ocean to a
hurricane, as compared to vertical mixing (Li and Zhong, 2007; Zhai et al., 2009; Dorostkar
et al., 2010). In the ocean basin of the present interest, the horizontal diffusion coefficient
was estimated to be an order of 10 $\mathrm{m^2/s}$ under extreme conditions, e.g., TC condition
(Allahdadi, 2014; Mulligan and Hanson, 2016). Thus, we take $\nu' = \kappa' = 10\,\mathrm{m^2/s}$ in the
present study for simplicity to simulate the ocean response to Hurricane Irene.
## 2.2 Computational Conditions

In order to fully capture the NIC induced by Hurricane Irene (2011), our computational

domain covers the entire MAB regions of the U. S. East Coast extending from Cape Cod,
Massachusetts, to Cape Lookout, North Caroline. The computational domain is discretized
into 35 layers with refinement near the surface and covered with a 5 km×5 km grid in the
horizontal plane. The 1 arc-min bathymetry data is obtained from ETOPO1 Global Relief
Model (Amante and Eakins, 2009) and resampled to a resolution of 5 km. The simulation
starts from 20 August, one week before the hurricane event and lasted for a period of 16
days. The time step is set to 1 min.

The initial and open boundary conditions of the seawater temperature and salinity, the

ocean flow velocities and the sea surface elevation are all from the Hybrid Coordinate Ocean
Model (HYCOM, https://www.hycom.org/) with a resolution of 1/12° in space and 3 hr in
time (Cummings, 2005; Chassignet et al., 2007). The initial stratification in the HYCOM is
examined through a comparison with the 4D data provided by Experimental System for
Predicting Shelf and Slope Optics (ESPreSSO, http://www.myroms.org/espresso/). Seven
tidal constitutes (M2, S2, N2, K2, O1, K1, Q1) included in the simulation are derived from
the ADvanced CIRCulation model (ADCIRC, https://adcirc.org/). Daily inflows from the
eleven largest rivers, containing Susquehanna River, Delaware River, Hudson River,





Potomac River, etc., are obtained from the United States Geological Survey (USGS,
https://waterdata.usgs.gov/). The so-called radiation-nudging condition is adopted at the
open boundaries (Marchesiello et al., 2001). Wet-and-dry option is activated at coastal
boundaries (Warner et al., 2013). The seabed boundary condition is required to satisfy:
$$\nu \frac{\partial \mathbf{u}}{\partial z} = \boldsymbol{\tau}_{b} = \rho \left[ \frac{\lambda}{\ln\left(\Delta z \,/\, z_{0}\right)} \right]^{2} \left| \mathbf{u}_{b} \right| \mathbf{u}_{b} \tag{7}$$

where, $\boldsymbol{\tau}_{b}$ is the bottom friction; $\lambda$ is the von Karman constant; $\mathbf{u}_{b}$ is the fluid velocity
at the center of the bottom layer; $\Delta z$ is the distance between the center of the bottom layer
and the seabed; $z_{0}$ is the bottom roughness, which is set to 0.02 m in MAB following
Churchill et al. (1994).

The hurricane wind forcing required in this study can be obtained from two sources,

i.e., the H*WIND data, with a spatial resolution of 6 km and a temporal resolution of 6 hr,
published by Atlantic Oceanographic and Meteorological Laboratory, National Oceanic and
Atmospheric Administration (AOML/NOAA) (https://www.aoml.noaa.gov/hrd/data_sub/
wind.html) (Powell et al., 1998) and the North American Mesoscale (NAM) data, with a
spatial resolution of 12 km and a temporal resolution of 3 hr, provided by National Centers
for Environmental Prediction (NCEP) (https://www.ncdc.noaa.gov/data-access/model-
data/model-datasets/north-american-mesoscale-forecast-system-nam) (Janjic et al., 2004).
In our computation, the former is used between 26 and 31 August (during the hurricane
event) because it has a better accuracy in capturing the maximum wind speed, while the
latter is used during other periods of the simulation. Reanalysis data for other atmospheric
forcing, such as the surface air temperature, air pressure, relative humidity, radiation and
precipitation are also available from NAM for determining the surface buoyancy fluxes. In
particular, the momentum flux $\tau_{s}$ can be estimated through (Fairall et al., 1996):
$$\tau_{s} = \rho_{a} C_{d} \, u_{10}{}^{2} \tag{8}$$

where, $\rho_{a}$ is the density of the air; $C_{d}$ is the coefficient for momentum; $u_{10}$ is the
horizontal wind speed at the 10-m level; Several studies have confirmed that $C_{d}$ does not





increase but level off or even decrease at high wind speeds (Emanuel, 1995; Powell et al.,
2003; Donelan et al., 2004). Besides, $C_d$ could be altered due to wave deformation in
response to bathymetry change, especially in coastal regions (Chen et al., 2018; Xu and Yu,
2021). Therefore, we choose a well verified formula based on the numerical results from
the improved wave boundary layer model presented by Chen and Yu (2016), Chen et al.
(2018) and Xu and Yu (2021), which considered the effects of both bathymetry and wind
speed to determine $C_d$:

$$C_d = C_{dw} + \frac{C_{d0} - C_{dw}}{(W_0 - W)^2}(u_{10} - W)^2 \qquad (9)$$

where $C_{d0}$ is a threshold value set to 0.001 for the wind stress at $u_{10} \leq W_0 = 5$ m/s, $C_{dw}$
is the saturated wind stress coefficient and $W$ is the saturation wind speed. We have

$$C_{dw} = \begin{cases} -1.86\times10^{-4}\ln\frac{gD}{W_D}+0.0025 & \frac{gD}{W_D} \leq 3 \\ 0.00225 & \frac{gD}{W_D} > 3 \end{cases} \qquad (10)$$

$$W = \begin{cases} 4.64\ln(\frac{gD}{W_D})+42.6 & \frac{gD}{W_D} \leq 0.6 \\ W_D & \frac{gD}{W_D} > 0.6 \end{cases} \qquad (11)$$

where $W_D$ set to 40 m/s is the saturation wind speed in deep water. Except for the
momentum flux, other air-sea fluxes, e.g., the sensible heat flux and the latent heat flux, are
determined based on the conventional bulk parameterization scheme (see Fairall et al. (1996)
for detailed description).

## 3 Ocean Responses to Hurricane Irene

3.1 Effect of hurricane on ocean surface flow
As shown in Figure 1, Hurricane Irene (2011) entered the Mid-Atlantic Bight (MAB)
area of the present interest at Cape Lookout, North Carolina as a Category-1 event at 12:00,
27 August, 2011 (UTC time, the same below) with a maximum sustained wind (MSW) of



over 38 m/s. It continued to move northeastward and made a landfall at Atlantic City, New
Jersey at 9:35, 28 August with a MSW of around 30 m/s. During its motion in the MAB area
of our interest, the radius of the hurricane wind field (the area with wind speed ≥32.9 m/s)
reached a large value of 140 km (Avila and Cangialosi, 2011).

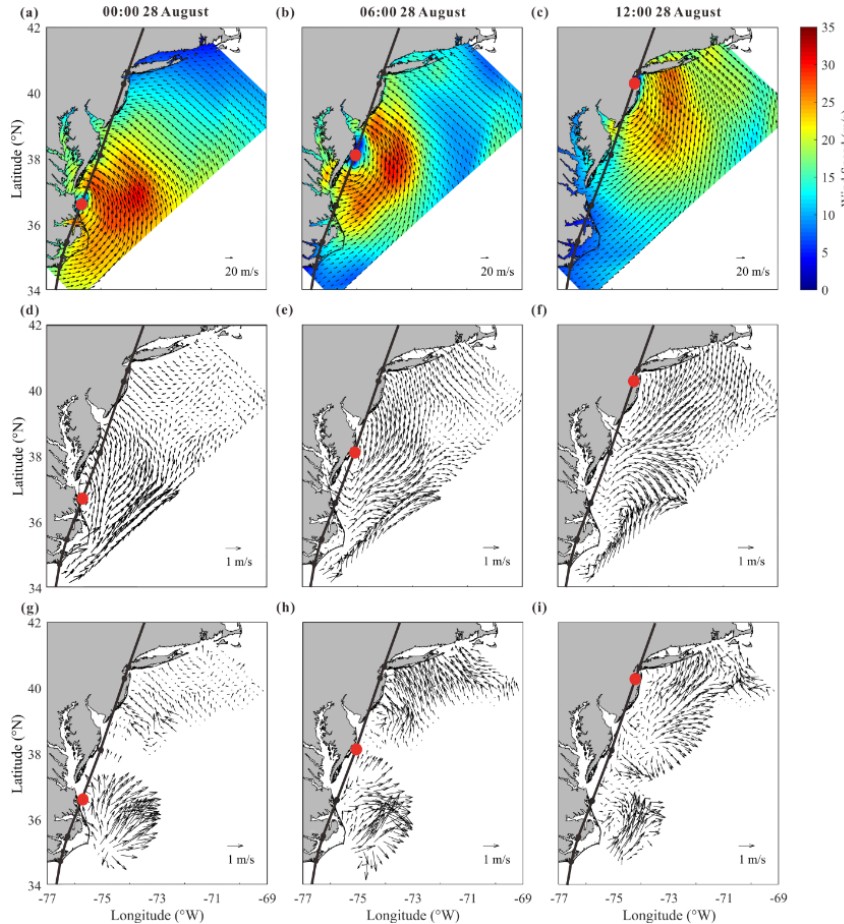


Figure 2. Snapshots of (a-c) the 10-m wind provided by H*WIND, (d-f) computed current
velocity of the surface layer and (g-i) observed current velocity of the surface layer, at (left
column) 00:00, (middle column) 06:00 and (right column) 12:00, 28 August, during the
passage of Hurricane Irene (2011). Note that best track of the hurricane reported by Avila
and Cangialosi (2011) is shown by black lines while the hurricane center is shown by red
circles.





Figure 2 provides the snapshots of the wind, the computed and observed currents in
the MAB area at 00:00, 06:00 and 12:00, 28 August, 2011, respectively. Note that 00:00 and
12:00 correspond to the time when Hurricane Irene entered and left the area of our interest,
respectively. The wind field is plotted from the H*WIND data. Field currents are obtained
by analyzing data from a network of High Frequency Radar (HF Radar) stations (Roarty et
al., 2010) in the Mid-Atlantic Regional Association's Coastal Ocean Observing System
(MARACOOS, https://maracoos.org/). The field data have a temporal resolution of 1 hr and
a spatial resolution of 6 km, and are assumed to be measured at an effective depth of 2.4 m
below the ocean surface. The data cover the MAB area from the coast to the shelf break and
demonstrate a reasonably good accuracy when compared to data obtained with ADCP
(Acoustic Doppler Current Profiler), which are usually considered to be reliable (Liu et al,

2014).

The computed current velocity of the surface layer, as shown in Figure 2d-f, is
compared with the observed one, as shown in Figure 2g-i, to verify the reliability of the
numerical model presented in this study. At 00:00, 28 August, it is numerically demonstrated
that currents rotating counterclockwise with a magnitude of over 1 m/s are rapidly generated
by the wind near the hurricane center (Figure 2d). In the observed results, though there are
significant data missing near the hurricane center, northeastward currents can still be
identified on the offshore waters along North Carolina coast (Figure 2g) and are in
reasonable agreement with the computed current field. Moreover, both computational and
observational results support a fact that the onshore wind (Figure 2a) on the front side of the
hurricane drives an onshore current with magnitude of 0.4 m/s along the northern MAB,
especially in the nearshore area of New Jersey (Figure 2d and 2g). At 06:00, Hurricane Irene
arrived at the offshore waters of Delmarva Peninsula. In spite of the field data missing, the
rotating currents induced by the hurricane wind can be clearly recognized in both computed
and observed results in the nearshore area of New Jersey (Figure 2e and 2h). In addition,
relatively strong onshore currents with magnitude of over 1 m/s are observed near Long
Island and are also well represented in the numerical results (Figure 2e). At 12:00, i.e., the



time when the hurricane left the area of our interest, the counterclockwise rotating currents
are still formed near the hurricane center as demonstrated by both computational and
observational results (Figure 2f and 2i). At the same time, clockwise rotating currents are
shown to be generated near Delmarva Peninsula in southern MAB after the hurricane passed
over. This fact is certainly confirmed by both computed and observed results, indicating
near inertial currents are activated after the hurricane event. Therefore, it becomes evident
that the rotating wind of the hurricane immediately forces a rotating current in the surface
layer of the ocean and induces an inertial current rotating in the opposite direction shortly
after the hurricane passed over. It is also worthwhile to emphasize that, in general, the
numerical results obtained with the present model agree fairly well with observed data.
## 3.2 Effect of hurricane on vertical stratification and sea surface cooling
Shown in Figure 3a is the vertical profile of the seawater temperature measured by
Glider RU16 launched off the New Jersey Coast. Glider RU16 was an autonomous
underwater vehicle of the Slocum glider platform developed by Teledyne-Webb Research
(Schofield et al., 2007, 2010), which has demonstrated to be advantageous in marine
monitoring, particularly under extreme weather conditions (Glenn et al., 2016; Miles et al.,
2017; Seroka et al., 2016; Zhang et al., 2018). Glider RU16 can measure not only the vertical
profiles of seawater temperature and salinity but also the water depth. During the hurricane
event, its position may include a certain amount of drift in the horizontal directions due to
the ambient flow.
In Figure 3a, it is seen that the mixed layer off New Jersey coast was quite thin, with a
thickness of less than 10 m, before the hurricane event. A strong stratification was clearly
formed over a water depth of 40 m, with a surface temperature of 24 °C and a bottom
temperature of 10 °C. When the hurricane center passed over the position of Glider RU16
at around 09:30, 28 August, the thickness of the mixed layer rapidly increased to nearly 30
m while the surface temperature was decreased by more than 5 °C, indicating a strong
mixing process has occurred. By plotting the time series of the squared buoyancy frequency



$N$ based on the measured data, expansion of the mixed layer due to the hurricane event
may be more vividly demonstrated (Figure 3c).

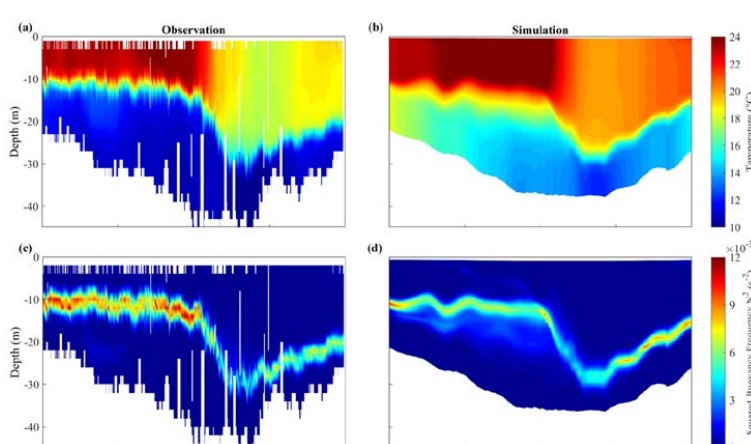


Figure 3. Time series of the vertical profiles of (top row) the temperature and (bottom row)
the squared buoyancy frequency, obtained from (a, c) Glider RU16 and (b, d) numerical
model.

Figure 3b and 3d present the computed results for the vertical distribution of seawater

temperature obtained by virtually setting a measuring point moving with the glider in the
real situation. The numerical results show a similar variation of the stratification pattern
before and during the hurricane event, indicating that the numerical model is capable of
describing the development and destruction of ocean stratification. However, a sea surface
cooling of about 4 °C obtained by the numerical model is a little smaller than 6-7 °C
observed by the glider in the field, probably due to the inaccurate setting of the initial bottom
temperature in the computation. In fact, the initial condition for the bottom temperature in
HYCOM is somehow higher (about 4°C) than the observed value in the field if Figures 3a
and 3b are compared. To correct this system error, the real-time profile obtained from RU16
is used for a nudging process in computation, i.e., the model temperature and salinity fields
are forced to nudge toward observed data (see Thyng et al. (2021) for detailed description).





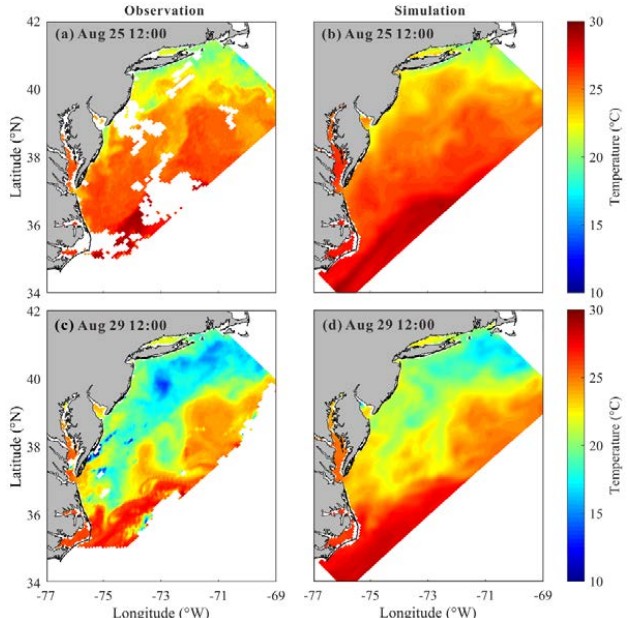

Figure 4. Sea surface temperature at Aug 25 12:00, before the hurricane event (top row) and at Aug 29 12:00, after the hurricane event (bottom row) from (a, c) observed data and (b, d) numerical model.

The sea surface temperatures (SST) before and after the hurricane event are further compared in Figure 4 (obtained from The Advanced Very High Resolution Radiometer (AVHRR), https://earth.esa.int/eogateway/catalog/avhrr-level-1b-local-area-coverage-imagery). Before the hurricane event, both observed and computed SST show similar patterns, i.e., the SST decreases with the increasing latitude. After the hurricane passage, the strong cooling mainly takes place in shallow waters, where the mixing is strong (Zhang et al., 2016), especially near New Jersey and Long Island. However, cooling is not prominent in shallow waters near North Caroline. In fact, the SST in this region had decreased and then recovered to its pre-hurricane level previously (Seroka et al., 2016). It should be pointed out that the computed SST cooling is 3-4 °C smaller than the observed one, which could also be explained by the inaccurate initial condition obtained from HYCOM. The HYCOM bottom temperature is somehow higher than actual, which could lead to the underestimation of the SST cooling. Therefore, we use the real-time SST data



obtained from AVHRR for nudging process in computation to correct this system error
(Thyng et al., 2021). Note that the error is mainly caused by the discrepancy in initial
settings but not the defects in numerical method. Thus this error could be calibrated in
certain extent and thus would not affect the reliability of subsequent analysis, e.g. energy
budget analysis.
## 3.3 Characteristics of NIC
To have a general understanding of the NICs in the MAB area induced by Hurricane
Irene (2011), a network of 30 stations aligned on 5 cross-shore sections from south to north
is introduced in this study to cover the area of our interest as shown in Figure 1a, similar to
Zhang et al (2018). In each section, 6 stations are placed in the cross-shore direction from
the shore side to the deep ocean, where water depths are around 30 m, 50 m, 75 m, 120 m,
and 220 m and 1000 m, respectively. Note that the most offshore stations are located outside
the shelf break.
The velocity of NIC is obtained from the total current velocity by first excluding the
tidal components and then passing it through a Butterworth filter with the frequency band
of 0.8-1.2 $f_0$, an effective approach proposed by Hormann et al. (2014), Zhang et al. (2018)
and Kawaguchi et al. (2020). Shown in Figure 5 are the time series of surface velocity of
the NIC component in the cross-shore direction at all stations during the time period of our
study (16 days from 20 August to 5 September). The alongshore component was similar to
cross-shore component and thus omitted here. It is demonstrated that the numerical results
are in reasonably good agreement with the HF Radar data and the Pearson product-moment
correlation coefficient reaches 0.7 (Derrick et al., 1994). The major sources of error in the
measured data are found at the most offshore stations, such as A6 and D6, where the
coverage of HF Radar is limited. In fact, Roarty et al. (2010) indicated that the observed
data outside the shelf break should be used with caution. Error in the numerical results of
the NICs may come from the minor errors in the wind forcing data because they are very
sensitively related, e.g., underestimation at C3-C6 before the hurricane event may come
from the errors in low-resolution NAM data used in pre-hurricane periods.



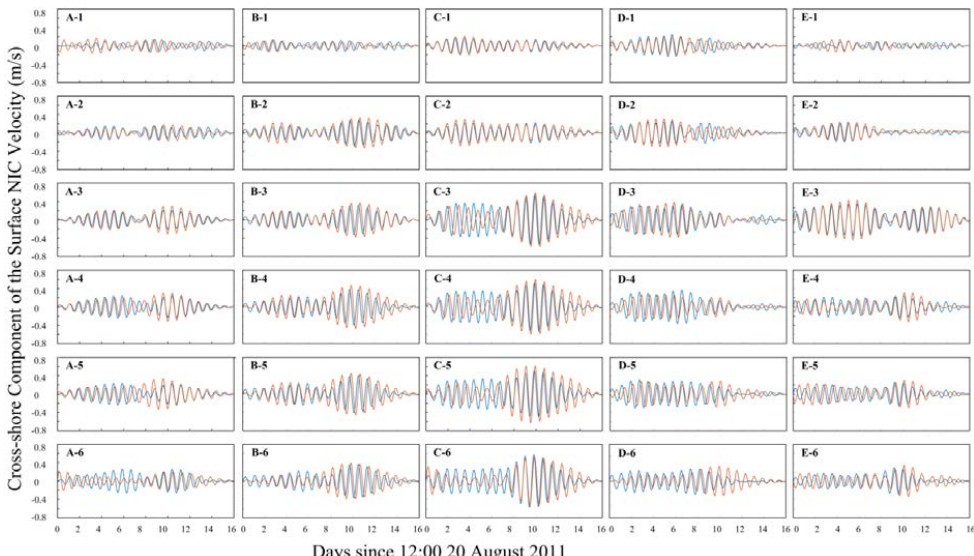

Figure 5. Time series of the NIC velocity in the surface layer obtained (blue line) by the HF Radar and (orange line) with numerical model at 30 stations along sections A-E.

In Figure 5, it can be readily recognized that, in the cross-shore direction from shallow to deep waters (i.e., Station No.1-No.5 in present study), the NIC velocity gradually increases by a factor of at least three, e.g., from 0.15 m/s to 0.6 m/s in section C, which is consistent with conclusions in previous studies (Kim and Kosro, 2013; Yang et al., 2015; Rayson et al., 2015; Zhang et al., 2018). This is because that NIC velocity in the nearshore region are restricted due to a combination of several reasons presented by Chen and Xie (1997), Rayson et al. (2015) and Chen et al. (2017). Different from other studies, however, the NIC velocity in the deep waters (i.e., Station No. 6 in the present study) is found to be not larger or even smaller than that nearby the shelf break. This is probably due to that fact that the track of Hurricane Irene (2011) was nearly attached to the shore during its motion in the area of our interest and the wind stress over the deep ocean was relatively small. From south to north, it is found that the NIC velocity in the middle regions, such as along section C, is larger than those in south and north. By checking the numerical results, it is found that the stratification was only slightly destroyed during the hurricane event near section C as



compared to the adjacent sections, which thus provided a better environment for NIC
generation (Yang et al., 2015; Shen et al., 2017).

To evaluate the relative importance of the near inertial currents, the rotary spectra of

the surface current velocity during the period of study (16 days) at different stations are
shown in Figure 6. The tidal flows corresponding to the major constituents M2, N2 and K1,
obtained with ADCIRC, are also plotted. It is seen that the velocity of the NICs is of an
equivalent magnitude to that of the M2 tidal current at the shallow-water stations where the
water depth is about 30 m (section C was taken for an example, Figure 6a). But, the velocity
of the NICs is significantly larger than that of the tidal current in deeper regions (Figure 6b,
c). It may be necessary to point out that weak NICs are not limited to the most nearshore
stations. In section D, for example, it is extended to a water depth of 75 m (Station D3,
Figure 6d). As discussed in the previous subsection, the weak NICs in the nearshore area
are closely related to the destruction of stratification by the strong mixing process associated
to the hurricane event (Yang et al., 2015; Shen et al., 2017). However, this effect does not
challenge the dominant role of NICs in deep waters.

Previous studies reported the nonlinear wave-wave interaction could transfer energy

from the M2 tide and NIC into a wave at the sum of their frequencies (fM2). The key
mechanism is the coupling between the vertical shear in NIC and the vertical velocity due
to the internal tide (Davis and Xing, 2003; Hopkins et al., 2014; Shen et al., 2017; Wu et al.,
2020). Though the M2 tide is rather strong in shallow waters during the hurricane event
(Figure 6), nonlinear wave-wave interaction between the tidal current and the NIC could be
hardly identified in most part of MAB. Nevertheless, a peak of the energy spectrum seems
to appear at the sum-frequency fM2 for the surface velocity at Stations B1 to B4, near
Delmarva Peninsula (B2 and B4 were taken as examples in Figure 6e, f). The evolution of
energy power at different frequencies for the middle-layer averaged (i.e., 10-30 m) currents,
where the flow shear is concentrated, is further demonstrated based on wavelet analysis
(Station B4 was taken as an example in Figure 6g). A peak energy at the sum-frequency fM2
is clearly identified after the hurricane passage. In fact, the subsequent Section 4.2 in this





paper will show that the strongest shear is found in offshore waters between Delmarva
Peninsula and New Jersey, i.e., near sections B and C (Figure 8a). Besides, Brunner and
Lwiza (2020) indicated that the most prominent M2 tide in southern MAB is located off
Delmarva Peninsula (near section B), according to a long-term observed data. Therefore,
the vertical shear in NIC and the vertical velocity due to the M2 tide is more likely to be
coupled in this region (i.e., near section B). However, this interaction only occurs in limited
regions and thus would not influence the NIC evolution in most part of MAB.

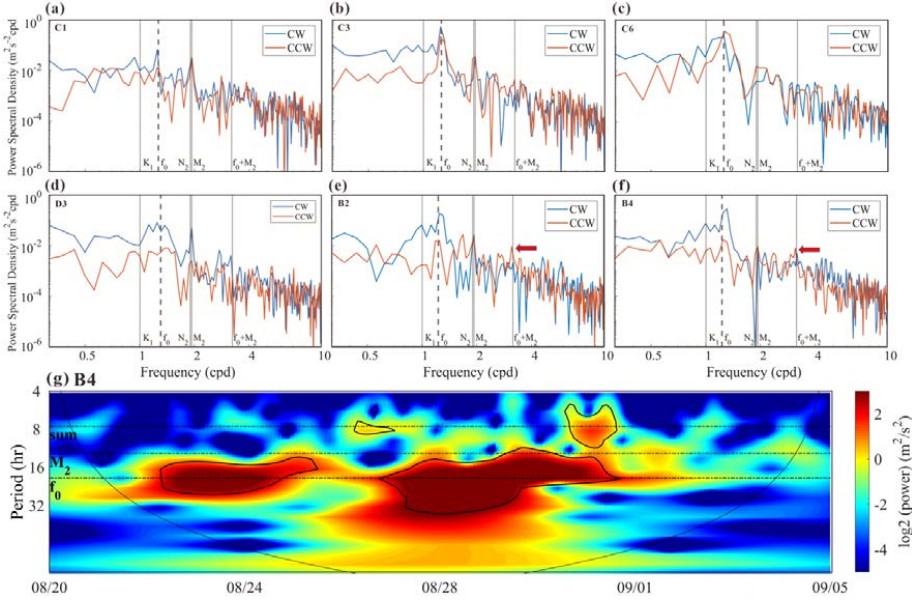


Figure 6. The rotary spectra of the current velocity in the surface layer during the simulation
time (16 days) obtained by HF Radar at Stations (a) C1 (~30m), (b) C3 (~75m), (c) C6
(~1000 m), (d) D3, (e) B2 and (f) B4. Clockwise and counter-clockwise components of the
current are shown by blue and orange lines, respectively (NICs are considered to be
dominated by the clockwise component). The frequencies of the major tidal constituents
M2, N2 and K1, the inertial frequency f0, and the sum-frequency of M2 and $f_0$ are all marked
by gray lines. (g) Wavelet power spectrum for 10-30 m depth-averaged alongshore current
component at Station B4 (see Thiebaut and Vennell (2010) for detailed description). Black
contours indicate the 5% significance level against red noise and the arc line indicate the
cone of influence.





## 4 Near Inertial Kinetic Energy

### 4.1 Conservation of NIKE

For description of the intensity of a NIC, the near inertial kinetic energy (NIKE) may be defined in the following way:

$$E' = \frac{1}{2}\rho_0 \left| \mathbf{u}' \right|^2 \tag{12}$$

where, $\mathbf{u}'$ is the velocity of the NIC; $\rho_0$ is the seawater density at the standard atmospheric pressure. Note that the NIKE is mainly gained from the wind power and dissipated due to a few mechanisms. Evolution of the vertically integrated NIKE within a water column from the sea bottom the ocean surface is thus governed by (Zhai et al., 2009)

$$\int_{-d}^{\eta} \frac{\partial E'}{\partial t} dz = \boldsymbol{\tau}_s \cdot \mathbf{u}_s' + \boldsymbol{\tau}_b \cdot \mathbf{u}_b' - \int_{-d}^{\eta} \rho_0 \nu_e \left| \frac{\partial \mathbf{u}'}{\partial z} \right|^2 dz - $$
$$\int_{-d}^{\eta} \nabla \cdot \left( \mathbf{u}'p' \right) dz - \int_{-d}^{\eta} \rho' g w' dz - \int_{-d}^{\eta} \nabla \cdot \left( \mathbf{U}E' \right) dz + \text{others} \tag{13}$$

where, $\mathbf{u}_s'$ and $\mathbf{u}_b'$ are near inertial velocities at sea surface and bottom, respectively; $\mathbf{U}$ is the sub-inertial velocity; $\rho'$ is the perturbation density, defined by $\rho' = \rho - \rho_*$; $\rho_*$ is the reference density, i.e., the density corresponding to a flattened stratification where the fluid is redistributed adiabatically to a stable and vertically uniform state from the actual condition (Holliday and McIntyre, 1981; Kang and Fringer, 2010; MacCready and Giddings, 2016); $p'$ is the perturbation pressure, defined by $p' = g\int_z^{\eta} \rho' dz$. Terms on the right-hand side of Eq. (13) are the wind energy input, the dissipation due to bottom friction, the vertical diffusion due to turbulence, the horizontal divergence of near inertial energy flux, the conversion between kinetic and potential energy, and the advection of NIKE by the sub-inertial flow. The last term 'others' includes nonlinear transfer of energy between NICs and flows of other frequencies as well as the horizontal diffusion due to mixing. Note that the energy are integrated over the water column from $z = -d$ to free surface $z = \eta$. In shallow waters, $d$ is the actual water depth, while in deep waters, $d$ is truncated to 200 m (i.e., the depth of the shelf break). When the bottom boundary is set at $z = -200\,\mathrm{m}$,




the bottom friction vanishes in Eq. (13) but a term related to the downward energy flux, i.e.,
$p'w'\big|_{z=-200m}$ should be added.
For a general understanding, distribution of the depth-integrated NIKE averaged over
a 10-day period from August 25 to September 4 is presented in Figure 7a. The wind power
integrated over the same period is plotted in Figure 7b. It is clearly shown in Figure 7a that
the high NIKE region mainly located in the offshore waters of Delmarva Peninsula and New
Jersey rather than in the nearshore area. This distribution pattern is rather similar to that of
the wind energy input, as presented in Figure 7b, indicating that the NIKE was immediately
gained from the wind power (Rayson et al. 2015; Shen et al., 2017; Zhang et al., 2018). In
fact, the NIKE could also come from other processes apart from the wind energy input
(Alford et al, 2016), meanwhile the wind energy input may also be transferred to energy of
waves apart from NIC (Chen at el., 2017), which leads to differences between Figure 7a and
7b.

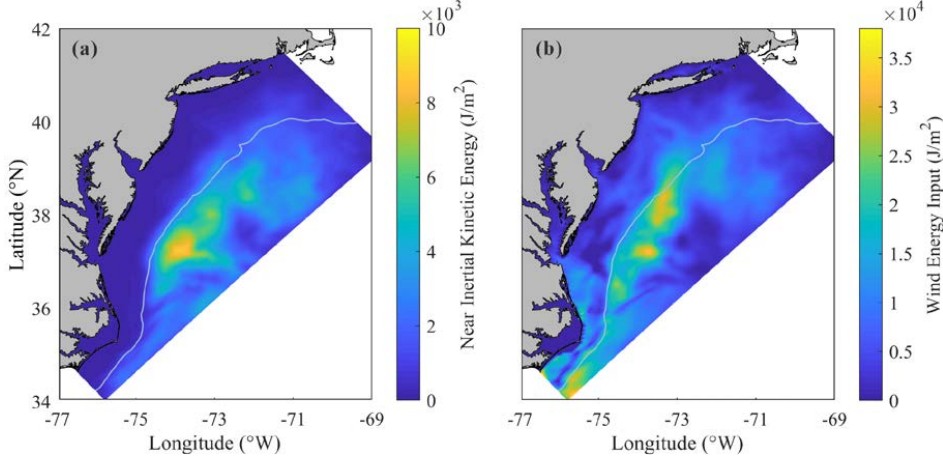

Figure 7. Spatial distribution of (a) depth-integrated near inertial kinetic energy averaged
over the 10-day period and (b) wind power input to NICs integrated over the 10-day period.






Table 1. The contribution of each mechanism to energy budget. Percentages in parentheses
refer to the ratio of each factor to wind energy input.

| Factor (J) | Contribution in Region A | Contribution in Region B |
|---|---|---|
| Wind Energy Input | $7.75 \times 10^{14}$ | $3.16 \times 10^{14}$ |
| Vertical Turbulence Diffusion | $3.12 \times 10^{14}$ (40%) | $2.12 \times 10^{14}$ (67%) |
| Lateral Divergence | $1.34 \times 10^{14}$ (17%) | $5.69 \times 10^{13}$ (18%) |
| Downward Transfer | $2.58 \times 10^{14}$ (33%) | 0 |
| Advection | $3.33 \times 10^{13}$ (4%) | $1.04 \times 10^{13}$ (3%) |
| Conversion | $6.9 \times 10^{12}$ (1%) | $1.58 \times 10^{13}$ (5%) |
| Bottom Friction | 0 | $7.58 \times 10^{13}$ (24%) |


An important objective of the present study is to identify the mechanism of NIC
development and decay. For this purpose, we consider a rectangular domain and separate it
into deep water region A (depth > 200 m) and continental shelf region B (depth $\leqslant$ 200 m),
as depicted in Figure 1a. If the NICs are considered to be negligibly weak before and after
Hurricane Irene (2011), we may try to find how the wind power that drives the NICs during
the hurricane event is balanced, by comparing the accumulated contribution of different
mechanisms. Performing an integration of each terms in Eq. (13) with respect to time over
10 days from August 25 to September 4 and with respect to the horizontal coordinates over
both deep water region A and continental shelf region B, the contribution of each mechanism
to the energy budget is obtained as shown in Table 1. It is clearly demonstrated that in the
deep water region, the wind energy input was basically balanced by the vertical diffusion
due to turbulence (40%) and a downward transfer of the near inertial energy to the deep
ocean (33%). In the continental shelf region, the vertical diffusion due to turbulence
dominated the dissipation of NIKE (nearly 70%), while the bottom friction played a
secondary role (24%). It is worthwhile mentioning that lateral divergence of NIKE should
not be neglected in both shallow and deep water regions under the hurricane condition




(nearly 20%), different from previous studies which focused on NICs under the local wind
condition or in a broader research region across the whole North Atlantic (Chant, 2001; Zhai
et al., 2009; Shen et al., 2017). Other processes, e.g., advection due to sub-inertial flows,
only played a minor role. Note that the ratio of near inertial energy decay to wind energy
input exceeded 100% in the continental shelf region, confirming that NIKE may be gained
from other sources in addition to wind energy input in nearshore regions (Alford et al., 2016).

## 4.2 Decay of NIKE

The spatial distribution of the time-integrated energy dissipated through vertical
diffusion due to turbulence is plotted in Figure 8a. It is seen that a large amount of the
dissipation occurred at the offshore side of the continental shelf (i.e., at the offshore side of
the shallow region B), which does not coincide with the region where the wind energy input
is intense as demonstrated in Figure 7b. This implies that dissipation of NIKE is not mainly
caused by an increased intensity of turbulence, which certainly takes place in a region where
wind energy input achieves a high level (Zhai et al., 2009; Zhang et al., 2018). For a more
detailed discussion, the averaged eddy viscosity $\nu_e$ and the averaged vertical shear rate of
NIC $\left| \partial \mathbf{u}' / \partial z \right|^2$ during the period of our study are presented in Figure 8b and 8c. It is then
confirmed that the strong vertical shear also occurred at the outer half of the continental
shelf. The eddy viscosity, however, has a completely different distribution. In conclusion,
the vertical shear, known to be closely related to the ocean stratification (Shen et al., 2017),
plays a crucial role in the turbulence diffusion. It happened that one of the well-known
sharpest thermoclines in the world exists in the coastal water of MAB (Schofield et al., 2008;
Lentz, 2017). It may be necessary to emphasize that, although the stratification in the
shallowest water was totally destroyed during the hurricane event, as mentioned in Section
3, the seawater at the outer half of the continental shelf still partly maintained its
stratification.




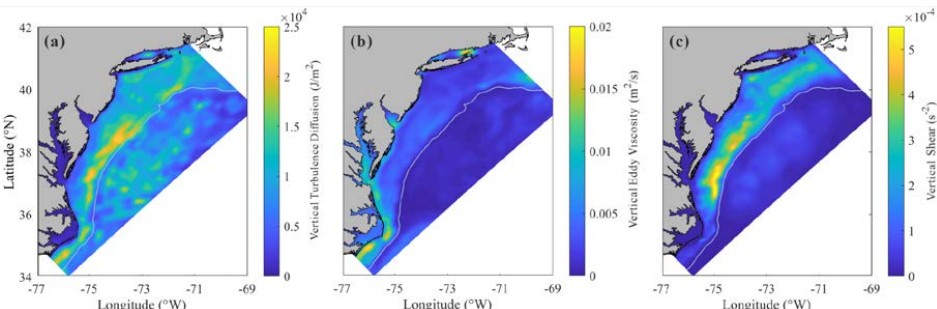


Figure 8. Spatial distribution of (a) depth-integrated vertical diffusion due to turbulence
integrated over the 10-day period, (b) depth-averaged vertical eddy viscosity and (c) depth-
averaged vertical shear, both averaged over the 10-day period.

The lateral divergence of NIKE flux, which also results in decay of NIKE and is not
trivial (~20%) in both shallow and deep water regions, may have to be discussed in some
details. As shown in Eq. (13), the lateral divergence of NIKE flux is a vertical integration
of $\nabla \cdot \left( \mathbf{u}' p' \right)$, which may also be expressed as an equivalent integration of $\nabla \cdot \left( \mathbf{c}' E' \right)$,
where $\mathbf{c}'$ is the transport velocity of NIKE in the horizontal plane (Price, 1994). When
compared to previous studies (Zhai et al., 2009), which dealt with the normal wind induced
NIC over a large part of the North Atlantic and showed that the lateral divergence accounted
only for less than 5% of the total NIKE loss, we focused only on the hurricane-affected
region. In the hurricane-affected region, the larger NIKE gradient naturally leads to a larger
divergence. If we extend the domain of study by a factor of 1.5, however, contribution of
the averaged lateral divergence decreases by more than half. It is thus strongly implied that
the lateral divergence of NIKE flux is significant within the hurricane-affected region.
It is also of interest to note that the contribution of the lateral divergence in south region
of our computational domain is less than 8%, much smaller than the average value of ~20%.
Several studies have pointed out that the transport velocity $\mathbf{c}'$ is largely influenced by the
background vorticity gradient (Zhai et al., 2009; Park et al., 2009). In other words, NIKE
can hardly be transferred from a place of lower background vorticity to a place of higher
background vorticity or, NIKE can hardly penetrate a vorticity ridge from either side. Shown
in Figure 9 is the distribution of the background vorticity within our computational domain



during the hurricane event (data from https://resources.marine.copernicus.eu/product-
detail/SEALEVEL_GLO_PHY_CLIMATE_L4_MY_008_057/INFORMATION). A
remarkable vorticity ridge exists in the southeast of the computational domain, which is
considered to be caused by the strong horizontal shear at the edge of Gulf Stream (a warm
and swift ocean current in Atlantic, flowing through the southern MAB and propagating
northeastward). This vorticity ridge can reduce the lateral divergence of NIKE flux in south
region of our computational domain.

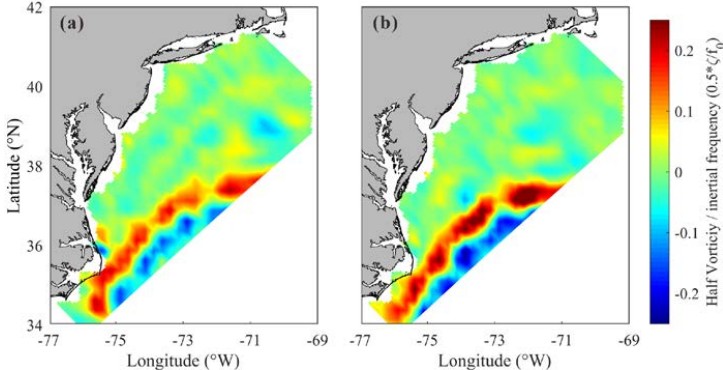


Figure 9. Spatial distribution of background vorticity (a) before the hurricane event on Aug
25 and (b) after the hurricane event on Sep 4.

4.3 Decay timescale of NIKE
It is of practical importance to determine the rate of NIKE decay. A conventional
measure of the rate of NIKE may be its e-folding time, i.e., the timescale in which the NIKE
decreases by a factor of e. Shown in Figure 10 is the e-folding time of the depth-integrated
NIKE at 24 stations along sections A to D. The decay timescale in section E is not considered
because this section is relatively far from the hurricane track as compared with other
sections and also because the orientation of section E differs quite significantly from that of
other sections.





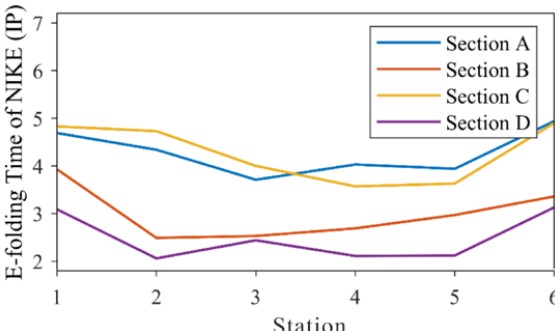

Figure 10. The decay timescale of the depth-integrated NIKE at 24 stations along sections A to D. Note that the unit for the e-folding time is the inertial period.

It is interesting to note that the decay timescales in the shallow and deep regions are fairly different. As shown in Figure 10, the NIKE is dissipated much more slowly outside the shelf break (Station No.6) than over the continental shelf. This difference is often considered to be an effect of the bottom friction and the extremely strong turbulence in the shallow waters, as pointed out by other researchers (Rayson et al., 2015; Shen et al., 2017). It is also interesting to find that the variation of NIKE decay rate in shallow waters is much more complicated than in the deep waters. In the cross-shore direction, the NIKE at the middle stations, i.e., Stations No. 3 to No. 5, located at the outer half of the continental shelf, is shown to be dissipated most rapidly, especially along sections A to C (Figure 10). This phenomenon is actually supported by the fact that the strongest turbulence diffusion occurred over the outer half of the continental shelf, particularly in the relevant region between sections A and C (Figure 8a). Considering the variation of the wind energy input within the same section should not be too large, the ratio of turbulence diffusion to wind energy input must be mainly determined by the turbulence diffusion. Therefore, the strong turbulence dissipation due to the strong vertical shear in well-maintained stratification is responsible for the rapid energy decay in the outer half of the continental shelf, as shown in Section 4.2. Although the bottom friction also has some effect on the decay timescale of NIKE onshore, the turbulence effect is predominant.



In the alongshore direction, it is shown that the NIKE in sections B and D decayed
more rapidly. Actually, the decay timescale there is only 2 to 3 inertial periods compared to
4 to 5 inertial periods in sections A and C. However, the limited variability of the turbulence
diffusion in alongshore direction should not lead to such a big difference. Near section A,
the vorticity ridge in Gulf Stream restricted the lateral divergence of NIKE, which may
contribute to a long decay timescale to some extent. However, the role of this effect was
limited. In fact, as mentioned in Section 3, the nonlinear wave-wave interaction near section
B may have caused a transfer of NIKE to other frequencies, as also pointed out by Shen et
al. (2017). In fact, it is found that the ratio of turbulence diffusion to wind input in section
B was larger than in other sections by 20%-30%, due to the low level of wind input (Figure
7b) and high level of turbulence dissipation (Figure 8a) there. These factors combined seem
to have yielded an extraordinarily short e-folding time in section B. In section D, due to the
complete destruction of stratification after the hurricane event (as mentioned in Section 3
and shown in Figure 6d), the NICs were of the same order as the background flow (D1-D4
in Figure 5). Therefore, the decay timescale of NIKE in section D is certainly inaccurate
and possibly meaningless.
**5 Conclusion**
This study is aimed to investigate the development and decay mechanism of NICs in
the MAB area caused by Hurricane Irene (2011). Numerical results obtained with ROMS
are shown to agree well with the observational data. Both computational and observational
results show that the rotating wind of the hurricane immediately forced a rotating current in
the surface layer of the ocean and induced an inertial current rotating in the opposite
direction about one inertial period after the hurricane passed over. The NICs overwhelmed
M2 tide in most areas of the MAB region except in the nearshore area where the
stratification was totally destroyed by the strong mixing due to turbulence. In addition, the
cross-shore component of the NIC velocity gradually increases by a factor of at least three
from a shallow-water position to the shelf break.



The energy budget in the NICs is investigated in both deep and shallow waters. NIKE
was shown to be immediately gained from the wind power during the hurricane event. In
the deep water region, NIKE was mainly dissipated by the vertical diffusion due to
turbulence and partially transferred to deep waters. In the continental shelf region, NIKE
was basically dissipated by the turbulence diffusion, meanwhile the bottom friction played
a secondary role. The nonlinear wave-wave interaction only dissipated NIKE in limited
regions, e.g. shelf waters off Delmarva Peninsula. Notably, the lateral divergence of NIKE
should be taken into consideration in both shallow and deep water regions under the
hurricane condition. However, in southern MAB, it was restricted by a vorticity ridge at the
edge of Gulf Stream. It is also clarified that the NIKE dissipation due to turbulence diffusion
is much more closely related to the rate of the vertical shear rather than the intensity of
turbulence, which certainly takes place in a region where wind energy input achieves a high
level. The strong vertical shear at the offshore side of the continental shelf leaded to the
strong turbulence dissipation in this region.
**Competing interests**
The authors declare that there is no conflict of interest.
**Authors' contributions**
P. Han and X. Yu conceived of the presented idea. P. Han performed the computations.
X. Yu supervised the project. Both authors discussed the results and contributed to the final
manuscript.
**Funding**
This research is supported by National Natural Science Foundation of China (NSFC)
under grant No. 11732008.



## Data Availability

The data used in this study are listed below. In particular, the regional oceanic modeling system (ROMS) code is available at https://www.myroms.org; HF radar data is available at http://tds.marine.rutgers.edu/thredds/dodsC/cool/codar/totals/5Mhz_6km_realtime_fmrc/ Maracoos_5MHz_6km_Totals-FMRC_best.ncd.html; Glider data is available at http://tds.marine.rutgers.edu/thredds/dodsC/cool/glider/mab/Gridded/20110810T1330_epa _ru16_active.nc.html; HYCOM data is available at https://www.hycom.org/data/glbu0pt08/ expt-91pt2; ADCIRC data is available at https://adcirc.org/products/adcirc-tidal-databases; USGS data is available at https://waterdata.usgs.gov; H*WIND data is available at https://www.aoml.noaa.gov/hrd/data_sub/wind.html; NAM data is available at https://www.ncdc.noaa.gov/data-access/model-data/model-datasets/north-american-mesoscale-forecast-system-nam; C3S data is available at https://resources.marine.copernicus.eu/product-detail/SEALEVEL_GLO_PHY _CLIMATE_L4_MY_008_057/INFORMATION; AVHRR data is available at https://earth.esa.int/eogateway/catalog/avhrr-level-1b-local-area-coverage-imagery.

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
