# Peer review of "A Numerical Study of Near Inertial Motions in Mid-Atlantic Bight Area Induced by Hurricane Irene (2011)"

_EGUsphere, 2022_

## Author Comment (AC1)

**Responses to Comments of Reviewer #1**

We appreciate very much the comments of Reviewer #1 and have revised the manuscript accordingly. In the following, we explain our response to the comments. The relevant revisions are highlighted with red color in the marked manuscript.

**Comments:**

A good and timely study about properties of inertial waves. What I miss is a more detailed description of the used boundary layer model (line 106) and how the definitions of its variables relate to the analysis. Related to that is the rather poor Figure 5. Getting observed and modelled NIWs right requires good forcing and a good ML model. So please highlight and enlarge important panels of Fig 5, come up with a metric (e.g., difference in NI EKE), and discuss differences - if any.

**Response:**

First of all, we would like to express our sincere thanks to the reviewer for his/her constructive comments on our study. We are very pleased to learn that the reviewer considers our study being "a good and timely" one about properties of inertial waves.

In this study, we used the regional oceanic modeling system (ROMS) (Shchepetkin and McWilliams, 2005) to compute the near inertial currents. We discretized the whole depth into 35 layers in the vertical direction and refined the near-surface layers. The sea surface boundary condition is required to satisfy:

$$\nu \frac{\partial \mathbf{u}}{\partial z} = \boldsymbol{\tau}_{\mathbf{s}} \tag{1}$$

where, $\nu$ is the viscosity of seawater, which was determined by the conventional k-ε turbulence model (Rodi, 1987; Umlauf and Burchard, 2003); $\tau_s$ is the wind drag given by (Fairall et al., 1996):

$$\tau_s = \rho_a C_d \, u_{10}{}^2 \tag{2}$$

where, $\rho_a$ is the density of the air; $C_d$ is the wind drag coefficient; $u_{10}$ is the horizontal wind speed at the 10-m level. To determine $C_d$, we preferred a formula that fits the numerical results obtained under extreme wind conditions with an improved wave boundary layer model

(Chen and Yu, 2016; Chen et al., 2018; Xu and Yu, 2021). So, the wave boundary model was not directly applied. The computed surface currents **u** is actually the averaged horizontal flow velocity within the top layer. The relevant modification is added in the revised manuscript [P8, L191-213].

We have improved the resolution of Figures 5 and 6, and enlarge the important panels [P16, Figure 5; P18, Figure 6]. We also introduced a metric, i.e., the classic Pearson product-moment correlation coefficient (Derrick et al., 1994), to verify the model:

$$r = \frac{\sum_{i=1}^{n}(X_i - \bar{X})(Y_i - \bar{Y})}{\sqrt{\sum_{i=1}^{n}(X_i - \bar{X})^2}\sqrt{\sum_{i=1}^{n}(Y_i - \bar{Y})^2}} \tag{3}$$

where r is the correlation coefficient, X and Y are the computed and observed results. The correlation coefficient reaches 0.7 in this study. It is thus concluded that the numerical results are in reasonably good agreement with the HF Radar data. The relevant modifications have been added in the revised manuscript [P17, L375-377].

**References:**

Chen, Y. and Yu, X., 2016. Enhancement of wind stress evaluation method under storm conditions. Climate Dynamics, 47(12): 3833-3843.

Chen, Y., Zhang, F., Green, B.W. and Yu, X., 2018. Impacts of ocean cooling and reduced wind drag on hurricane katrina (2005) based on numerical simulations. Monthly Weather Review, 146(1).

Derrick, T.R., Bates, B.T. and Dufek, J.S., 1994. Evaluation of time-series data sets using the pearson product-moment correlation coefficient. Medicine and science in sports and exercise, 26(7): 919-928.

Fairall, C.W., Bradley, E.F., Rogers, D.P., Edson, J.B. and Young, G.S., 1996. Bulk parameterization of air-sea fluxes for tropical ocean-global atmosphere coupled-ocean atmosphere response experiment. Journal of Geophysical Research: Oceans, 101(C2): 3747-3764.

Freeman, N.G., Hale, A.M. and Danard, M.B., 1972. A modified sigma equations' approach to the numerical modeling of great lakes hydrodynamics. Journal of Geophysical Research 77(6): 1050-1060.

Rodi, W., 1987. Examples of calculation methods for flow and mixing in stratified fluids. Journal of Geophysical Research: Oceans, 92(C5): 5305-5328.

Umlauf, L. and Burchard, H., 2003. A generic length-scale equation for geophysical turbulence models. Journal of Marine Research, 61(2): 235-265.

Shchepetkin, A.F. and McWilliams, J.C., 2005. The regional oceanic modeling system (roms): A split-explicit, free-surface, topography-following-coordinate oceanic model. Ocean Modelling, 9(4): 347-404.

Xu, Y. and Yu, X., 2021. Enhanced atmospheric wave boundary layer model for evaluation of wind stress over waters of finite depth. Progress in Oceanography, 198: 102664.

---

## Author Comment (AC2)

**Responses to Comments of Reviewer #2**

We appreciate very much the comments of Reviewer #2 and have revised the manuscript accordingly. In the following, we explain our response to each comment in a question-and-answer format. The relevant revisions are highlighted with red color in the marked manuscript.

**General Comments:**

Overall an interesting paper on an important topic with a storm that has become a wonderful test case for coastal ocean storm interactions. The study is well formed and remains largely focused on storm induced inertial currents. Some minor additions and edits are required, including a more detailed and distinct methods section for the observational data utilized. While the data was generally publicly available, more details on how the authors treated the data for QAQC, or what default QAQC if any they used from the downloaded data is required.

**Response:**

First, we would like to express our sincere thanks to the reviewer for his/her constructive comments on our study. We are very pleased to learn that the reviewer consider our study focusing on "an interesting paper on an important topic". Specific comments are addressed in the following contexts.

**Detailed Comments:**

(1) Line 32 - 35 - while an interesting comment it is disconnected from the current article.

**Response:**

Thanks for the comment. We have revised this comment [P2, L32-35].

(2) Line 96 - Caroline should be Carolina.

**Response:**

We are sorry for the mistake. We have carefully checked the entire manuscript to avoid such mistakes.

(3) Line 99 - What was the vertical gradient in temperature? This is likely more important than the surface/bottom temperature difference.

**Response:**

Thanks for the comment. In fact, the vertical gradient in temperature was very large before the hurricane passage. Glenn et al. (2016) and Seroka et al. (2017) analyzed the glider data and indicated that the thermocline in MAB shelf region was quite thin, e.g. the thermocline was less than 5 m where the water depth was around 40 m. Considering that the surface/bottom temperature difference was larger than 10 °C, the vertical gradient in temperature within the thermocline could be large than 2 °C/m. More detailed explanation and the relevant reference is added in the revised manuscript [P5, L98-101].

(4) Line 102 - Schofield et al., 2010 is a reference for Slocum gliders generally, however, there are multiple references for the Hurricane Irene specifically (Glenn et al., 2016) being the most prominent. There is no clear methods/data section, with some of the observational data described within what looks like results sections.

**Response:**

Thanks for the kind suggestion. We add the relevant references for the Hurricane Irene (Glenn et al., 2016; Seroka et al., 2016; Seroka et al., 2017) in the revised manuscript [P10, L219-220]. In addition, an introduction of the observational data, e.g., glider and HF Radar, is provided in section 2.3 [P10, L214-237].

(5) Line 234 - Why is the effective depth assumed to be 2.4m? Is there a reference for this?

**Response:**

Thanks for the comment. Roarty et al. (2020) indicated that the effective depth of

the measurement could be estimated according to the frequency of the radar. The averaged depth was estimated to be around 2.4 by Zhang et al. (2018) and to be around 2.7 m by a more recent publication, Roarty et al. (2020). A comment and the relevant references are added in the revised manuscript [P10, L223-225].

(6) Line 236 - Is the accuracy of HF Radar here referring to this dataset in particular or more generally? A more recent publication from Roarty et al., on HF Radar in the region can be found here:
https://agupubs.onlinelibrary.wiley.com/doi/full/10.1029/2020JC016368.

**Response:**

Thanks for the comment. The accuracy of HF Radar here is referring to the general dataset. Following the reviewer's suggestion, we have carefully studied Roarty's publication. Roarty et al. (2020) showed a more detailed description of the accuracy of HF Radar. The RMS differences of HF Radar data were within 8 cm/s when compared with ADCP (Roarty et al., 2010; Roarty et al., 2020). Comments and references are accordingly modified in the revised manuscript [P10, L225-227].

(7) Line 239 - 264 - Were tides removed from the HF Radar fields and model current fields? I believe later in the paper they were, but it's not clear what was done for this spatial analysis.

**Response:**

Thanks for the comments. We are sorry for not describing it clearly. In fact, the tides were removed from the HF Radar fields and model current fields for this spatial analysis. Relevant comments are added in the revised manuscript [P12, L259].

(8) Line 266 - 274 - Please comment on data QAQC, Glider setup details, or where this information can be found e.g. previous publications or where it was downloaded. I'm assuming it was from the IOOS Glider DAC? Also an additional paper on Hurricane Irene mixing from glider and ROMS data was detailed here https://agupubs.onlinelibrary.wiley.com/doi/full/10.1002/2017JC012756. And a

detailed exploration of pre-storm mixing was carried out by Watkins and Whitt here: https://journals.ametsoc.org/view/journals/phoc/50/12/jpo-d-20-0134.1.xml

**Response:**

Thanks for the suggestion. Glider RU16 was an autonomous underwater vehicle of the Rutgers Slocum glider platform developed by Teledyne-Webb Research. It was equipped with the Seabird un-pumped conductivity, temperature, and depth (CTD) sensor and launched in Aug 10, 2011. RU16 moved vertically through the water column and typically collected data every 2 s. It was programmed to surface at nearly 3 h intervals to provide high temporal resolution data. It could measure both the vertical profiles of seawater temperature and salinity (Schofield et al., 2007; Glenn et al., 2016; Seroka et al., 2016). The accuracy of the dataset was closely related to the quality of the equipped sensors. More detailed description could be found in previous studies which used RU16 (Glenn et al., 2016; Seroka et al., 2016) or the website of Rutgers Slocum glider platform (https://rucool.marine.rutgers.edu/data/underwater-gliders/). Relevant comments and references are added in the revised manuscript [P10, L228-237].

(9) Line 290 - 301 - How did the maximum N-squared values compare between the observations and glider? It appears in Figure 3c and 3d that the observed N-squared was significantly greater than in the model ahead of the deepening and mixing, but similar during the deepening event?

**Response:**

Thanks for the comment. It is known that the N-squared value was calculated based on the vertical gradient of the potential density anomaly. Within the thermocline, the vertical gradient of the potential density anomaly was very large due to the large gradient of temperature T and salinity S. Therefore, N-squared values were quite sensitive to the T and S in the thermocline. It also means that, ahead of the deepening and mixing, a small error of T and S may lead to a prominent discrepancy of N-squared values in the thermocline. However, during the deepening event, the thermocline was nearly destroyed and the small error of T and S would not lead to such discrepancy any more.

Though there are discrepancies between the computed and observed values, the overall comparison was more than enough to validate our model. Both computed and observed N-squared values clearly showed the expansion of the mixed layer due to the hurricane event and capture the mixing process in seawater. The comments are added in the revised manuscript [P14, L323-325].

(10) Line 314 - I'm not clear on the use of the Zhang reference here. Is this referring to tropical cyclone shallow water mixing generally?

**Response:**

Thanks for the comment. The hurricane generally lead to the strong mixing and cooling in shallow waters where the initial stratification is strong. More detailed comments are added in the revised manuscript [P15, L341-343].

(11) Line 315 - Caroline should be Carolina

**Response:**

We are sorry for the mistakes. A relevant correction has been made.

(12) Line 315 - 316 - Over what time-scale did the SST recover to pre-hurricane levels? Off North Carolina there was likely very little Cold Pool water, thus mixing should result in very little cooling? Plots of bottom. Temperature pre-storm from the model will likely show this.

**Response:**

Thanks for the valuable suggestion. In fact, the SST recovered to pre-hurricane levels within only 1 day (Seroka et al., 2016). As the review suggested, we checked the pre-storm temperature profile at Station A2 off North Carolina. As shown in the figure, the initial temperature difference between the surface and bottom is smaller than 10 °C, and the bottom temperature is as high as 18 °C. Thus, little Cold Pool water may lead to insignificant cooling and fast recovering as the reviewer supposed. A modification is made in the revised manuscript [P15, L344-348].

[Figure]

(13) Line 323 - I agree that the model/data mismatches are largely not too critical for the process investigations presented here. I think the strength of stratification is likely the most important model feature to validate as it can affect the vertical mixing and generation/dissipation of NIC.

**Response:**

Thanks for the kind suggestion. The relevant comments are added in the revised manuscript [P16, L354-355].

(14) Line 342 - 343 - Were data dropouts documented, or could there be dynamical reason that the NIC are in poorer agreement offshore? The HF Radar data should include quality flags to identify missing or low quality data.

**Response:**

Thanks for the comment. The HF Radar data at each point were recorded and integrated from several radars in the observational network. Therefore, the quality of the data was largely decided by the 'coverage' of radars. Studies indicated that in shallow water regions, the coverage was larger than 90%. When compared with ADCP, the RMS difference of HF Radar was only within 8 cm/s in shallow water regions. However, the coverage dropped to ~50% outside the shelf break. Several studies have showed that data are unreliable and should be viewed with caution if the coverage is less than 50% (Roarty et al., 2010; Kohut et al. 2012; Roarty et al. 2020). Therefore, the low coverage should be the culprit causing the poor agreement outside the break. The relevant comments and references are added in the revised manuscript [P17, L378-382].

(15) Line 378 -381 - Is the 75m D3 location the beginning of the shelf-break front, a mesoscale feature impinging on the shelf, or simply too far from the main track? Adding the reference lines to additional spatial figures would be helpful doer interpretation rather than needing to flip back to figure 1 for the reader.

**Response:**

Thanks for the comment. In section D, NICs were quite weak from the shore to D3 due to the destruction of stratification in nearshore regions. However, the NICs were prominent outside D3. Because the stratification outside D3 was relatively well maintained due to the thicker mixed layer in these regions and the farther distance from the main hurricane track. An additional figure with the reference lines is added in the revised manuscript [P16, Figure 5].

**References:**

Glenn, S.M. et al., 2016. Stratified coastal ocean interactions with tropical cyclones. Nature Communications, 7(1): 10887.

Kohut, J., Roarty, H., Randall-Goodwin, E., Glenn, S. and Lichtenwalner, C.S., 2012. Evaluation of two algorithms for a network of coastal HF Radars in the mid-atlantic bight. Ocean Dynamics, 62(6): 953-968.

Roarty, H. et al., 2010. Operation and application of a regional high-frequency radar network in the Mid-Atlantic Bight. Marine Technology Society Journal, 44(6): 133-145.

Roarty, H. et al., 2020. Annual and seasonal surface circulation over the mid-atlantic bight continental shelf derived from a decade of high frequency radar observations. Journal of Geophysical Research: Oceans, 125(11): e2020JC016368.

Schofield, O. et al., 2007. Slocum gliders: Robust and ready. Journal of Field Robotics, 24(6): 473-485.

Seroka, G. et al., 2016. Hurricane Irene sensitivity to stratified coastal ocean cooling. Monthly Weather Review, 144(9): 3507-3530.

Seroka, G. et al., 2017. Rapid shelf-wide cooling response of a stratified coastal ocean to hurricanes. Journal of Geophysical Research: Oceans, 122(6): 4845-4867.

Zhang, F., Li, M. and Miles, T., 2018. Generation of near-inertial currents on the Mid-Atlantic Bight by hurricane Arthur (2014). Journal of Geophysical Research: Oceans, 123(4): 3100-3116.

---

## Author Response (AR2)

**Responses to Comments of the Topic Editor**

We appreciate very much the comments of the topic editor and have revised the manuscript accordingly. In the following, we explain our response to each comment in a question-and-answer format. The relevant revisions are highlighted with red color in the marked manuscript.

**General Comments:**

In general, I did not find your responses sufficiently convincing. In places your responses to the reviewers were detailed but the text added to the paper was minimal. Please ensure that information that the reviewers asked for is added to the paper.

**Response:**

Many thanks to the topic editor's comments for improving the manuscript. We are sorry for the insufficient previous revision and we tried to add more information in this further revised version.

**Detailed Comments:**

(1) Please could you revise Figures 3, 4, 7, and 10 to use a colour-blind-friendly colormap, i.e. not colormap jet in Matlab. There are plenty of colormaps available now that are more accessible to all. See for example this article from 2014 https://www.climate-lab-book.ac.uk/2014/end-of-the-rainbow/.

**Response:**

Thanks for the kind suggestion. We have revised Figures 2, 3, 4, 7, and 10 accordingly [P12, Fig 2; P14, Fig 3; P16, Fig 4; P21, Fig 7; P27, Fig 10].

(2) Reviewer 1 asks for more information on the boundary layer (mixed layer) formulation that you use in ROMS. I do not think you have really answered this with the information provided in the paper. Your response is to repeat more detail of ROMS. What the reviewer is asking for is not to put information in the response, but to put information into the paper, exactly how the upper ocean boundary layer is treated in

**Response:**

We are sorry for the unclear description of our numerical method. In fact, the boundary layer effect on the near inertial current is not directly considered in this study. In this sense, we do not really have a boundary layer formulation in the present study.

In this study, the driving effect of the airflow on the near inertial current is realized by adding a wind drag on the ocean surface. Traditionally, the wind drag may be determined by an empirical formula that includes a drag coefficient and the drag coefficient is expressed as a linear function of the wind speed. In this study, we adopted a more advanced formula that fits the numerical results obtained under extreme wind conditions. The numerical results were obtained with an improved wave boundary layer model and reported previously (Chen and Yu, 2016; Chen et al., 2018; Xu and Yu, 2021). Since the wave boundary layer is not directly applied in the present study, we think that it may be more reasonable not to include the detailed formulation but to add a statement with supporting references. The relevant statement is added in the revised manuscript [P8, L190-192; P9, L196-199].

(3) Reviewer 1 commented that Figure 5 (now Figure 6) was "rather poor" and asked you to "please highlight and enlarge important panels of Fig 5". This figure looks the same to me. It needs improvement. Please highlight and enlarge important panels, and consider also plotting the differences between the lines.

**Response:**

We are sorry for our misunderstanding. We have highlighted and enlarged important panels in the revised manuscript [P19, Figure 6]. Please note that the apparently poor resolution of the figure is basically due to a limitation of MSWORD. We have provided a high-resolution version of the figure for typesetting when the paper is finally accepted for publication.

Since a very small discrepancy in the phase could lead to a large difference between the lines, we prefer not to plot the differences between the lines, to avoid misleading. Alternatively, we introduced a phase corrected parameter Δ to describe differences between observed and computed results (see our response to next comment). In fact, the phase of

NIC is quite sensitive to the initial wind data and a small phase discrepancy may not be a problem at all. The relevant discussion is added in the revised manuscript [P17, L381-400; P18, L403-404].

(4) Reviewer 1 also comments that you need a metric to determine how good the model is. You choose to use correlation. However two things could be perfectly correlated but orders of magnitude different. Therefore you need to come up with a different metric. I suggest you use the metric suggested by the reviewer - the difference in near-inertial EKE. You will need to add discussion of the differences in the paper.

**Response:**

Thanks for the suggestion. We totally agree with the reviewer. Actually, we have introduced a metric - the near inertial kinetic energy (NIKE) in Section 4.2, which is similar to the near-inertial EKE suggested by the reviewer. NIKE is advantageous because it could be used not only for verification of the NICs in Section 3.3 but also for verification of the subsequent results in Section 4.2. The near inertial kinetic energy (NIKE) is defined in the following way:

$$E' = \frac{1}{2} \rho_0 \left| \mathbf{u}' \right|^2$$

where, $\mathbf{u}'$ is the velocity of the NIC; $\rho_0$ is the seawater density at the standard atmospheric pressure.

Using the conventional relative mean square error to describe the difference in NIKE may lead to misunderstanding, since a very small discrepancy in the phase could lead to a large value of the mean square error. In fact, the phase of NIC is quite sensitive to the initial wind data and a small phase discrepancy may not be a problem at all. Therefore, we newly introduce a phase corrected relative mean square error:

$$\Delta = \frac{\min\limits_{\tau} \int_{t_0}^{t_1} \left[ E'_O(t) - E'_C(t-\tau) \right]^2 dt}{\sqrt{\int_{t_0}^{t_1} \left[ E'_O(t) \right]^2 dt} \sqrt{\int_{t_0}^{t_1} \left[ E'_C(t) \right]^2 dt}}$$

where $E'_O(t)$ and $E'_C(t)$ are the observed and computed NIKE time series; $[t_0, t_1]$ is the duration when the hurricane-induced NICs are prominent, which is taken to be from August 25 to September 4 in this study. $\tau$ is a time shift for eliminating the phase error.

We calculate $\Delta$ of the NIKE at all stations. It is shown that $\Delta$ varies from 0.14-0.23

in most stations. It is also necessary to mention that in several nearshore stations, i.e. A1, D1 and E1, $\Delta$ exceeds 0.3, because the NIC is too weak at these stations as compared to the background currents. At the 6 stations outside the shelf break, i.e., at A6, C6 and D6, $\Delta$ even exceeds 0.5-0.6, implying that the HF Radar data outside the shelf with low 'coverage' is less accurate. As we mentioned in Section 2.3, the relative RMS difference of HF Radar data is around 0.10. Taking this intrinsic HF Radar uncertainty into consideration, $\Delta = 0.14$-$0.23$ in our study is quite acceptable and reasonable. Therefore, we could conclude that our numerical results are in reasonably good agreement with the HF Radar data. The relevant discussion is added in the revised manuscript [P17, L381-400].

(5) Reviewer 2 asked that "While the data was generally publicly available, more details on how the authors treated the data for QAQC, or what default QAQC if any they used from the downloaded data is required." While you have added a section describing the data, thank you, you did not mention any quality control. Did you simply download other people's publicly available data and use them as they are? If so, you should say so clearly. Did you pay attention to any flags? Did you check the data for outliers etc? It is important to know exactly what *you* did. For example did you grid the data? Smooth or average them, in time and/or in space? You were not involved in the glider data acquisition or the HF radar acquisition I think, so you should make sure that those who did are clearly acknowledged and that you state in the text where and when you downloaded the data, what resolution or version you used, and what you did with the data?

**Response:**

Thanks for the comment. We are sorry for the insufficient statement of the QAQC process. We obtain the 1-hourly and 6-km HF Radar data from https://maracoos.org/ and use them as they are. The only thing we do is to interpolate the HF Radar data spatially to the 30 stations.

We did pay attention to the 'coverage' of the data we obtained, which is considered as a reliable quality flag. Generally, the surface current vectors provided by HF Radar are determined by combining overlapping radials from different radars in the observational network using an optimal interpolation method (Roarty et al., 2010; Zhang et al., 2018).

'Coverage' is directly related to how many overlapping radials are combined, and thus to the accuracy of data at a point. Previous studies pointed out that when the 'coverage' was larger than 90%, the data was rather reliable. When compared with ADCP, the RMS difference of HF Radar under such 'coverage' is only within 8 cm/s (Roarty et al., 2010; Kohut et al. 2012; Roarty et al. 2020).

We checked the HF Radar data used in our study. All of the data within the shelf break is quite reliable because the 'coverage' is larger than 90%. However, the data outside the shelf break has only a coverage of 60%-90%, because these regions are at the edge of the observational network. Though we present all of the data as they are, we must remind that the data outside the shelf break should be viewed with caution. The detailed description is added in the revised manuscript [P10, L221-236; P17, L377-380].

As for the glider data, we use RU16 conducted by Rutgers University. This dataset has been widely used and well verified in several previous studies (Glenn et al., 2016; Seroka et al., 2016; Seroka et al., 2017). Thus, we directly download their dataset from https://tds.marine.rutgers.edu/thredds/dodsC/cool/glider and use them as they are. The relevant description is added in the revised manuscript [P11, L245-248].

(6) Thank you for clarifying that you detided the model currents and the HF radar currents. Please can you state how you did this, with a reference if appropriate?
You added a mention of there being little Cold Pool water present, as the reviewer suggested. Could you expand on your text here please, to explain why little cold pool water leads to insignificant cooling and fast recovery?

**Response:**

Thanks for the comment. We used T-tide Matlab toolbox, a widely used and well validated tool (Pawlowicz et al., 2002), to detide the currents. The relevant reference is added in the revised manuscript [P12, L269-271].

Several studies showed that the sea surface cooling was positively related to the vertical temperature gradient in ocean (Shay and Brewster, 2010; Vincent et al., 2012, Zhang et al., 2016). They indicated that a small temperature difference between the surface and subsurface in ocean could lead to weak mixing effect and, hence, insignificant cooling and fast recovery. The description and relevant references are added to the revised manuscript [P15, L343-346].

(7) Reviewer 2 says that "the strength of stratification is likely the most important model feature to validate", but you have not validated this? Please could you quantify how well the model reproduces the observed strength of stratification, and add a discussion of this.

**Response:**

Thanks for the comment. We have simulated both the variation of temperature profile and the variation of mixed layer thickness during the hurricane event. By comparing them with RU16 data, we verify that our numerical model is capable of describing the development and destruction of ocean stratification. We add more discussions in the manuscript [P14, L300-302; P15, L319-322].

(8) You have not really responded sufficiently to "Were data dropouts documented, or could there be dynamical reason that the NIC are in poorer agreement offshore? The HF Radar data should include quality flags to identify missing or low quality data.". The question needs to be addressed - did you use the quality flags? You have responded with information about coverage, but what does this mean? Why was there low coverage? Or if you are simply downloading someone else's data, what do the quality flags mean and how did you use them? Are there data dropouts? Are there dynamical reasons for poor agreement? This needs discussion in the paper, beyond the sentence you added.

**Response:**

Thanks for the comment. We are sorry for the insufficient explanation about the quality flag of the data. We use 'coverage' as a flag to represent the quality of the HF Radar data. HF Radar measures the radial component of ocean surface currents using the Doppler Effect. The surface current vectors are determined by combining overlapping radials from different radars in the observational network based on an optimal interpolation method (Roarty et al., 2010; Zhang et al., 2017). Therefore, the 'coverage' of the radars indicates how many overlapping radials from different radars are combined. It is thus closely related to the accuracy of data at a fixed point. Previous studies reported that the data are quite reliable when the 'coverage' was larger than 90%. When compared with ADCP, the RMS

difference of HF Radar is only within 8 cm/s under such 'coverage' (Roarty et al., 2010; Kohut et al. 2012; Roarty et al. 2020).

In this study, we carefully checked the data quality. All of the data within the shelf break are well qualified since the 'coverage' is larger than 90%. However, the data outside the shelf break only has a coverage of 60%-90%. In fact, we use all of the data as they are, however, we must remind that the data outside the shelf break should be viewed with caution. In particular, a possible poor agreement between simulated and observed NICs at Stations A6-E6 does not necessarily mean a low accuracy of the computational results. The description is added in the revised manuscript [P10, L221-236; P18, L395-397].

(9) Thank you for adding the extra figure showing the lines. However you have not added anything to the paper in response to "Is the 75m D3 location the beginning of the shelf-break front, a mesoscale feature impinging on the shelf, or simply too far from the main track?". Please add some sentences to address this.

**Response:**

Thanks for the comment. In section D, NICs were quite weak from the shore to D3 due to the destruction of stratification in nearshore regions. However, the stratification outside D3 was relatively well maintained due to the thicker mixed layer in these regions and the farther distance from the main hurricane track, as the reviewer commented. We have added the description into the revised manuscript. [P20, L434-436].

**References:**

Chen, Y. and Yu, X., 2016. Enhancement of wind stress evaluation method under storm conditions. Climate Dynamics, 47(12): 3833-3843.

Chen, Y., Zhang, F., Green, B.W. and Yu, X., 2018. Impacts of ocean cooling and reduced wind drag on hurricane katrina (2005) based on numerical simulations. Monthly Weather Review, 146(1).

Glenn, S.M. et al., 2016. Stratified coastal ocean interactions with tropical cyclones. Nature Communications, 7(1): 10887.

Kohut, J., Roarty, H., Randall-Goodwin, E., Glenn, S. and Lichtenwalner, C.S., 2012. Evaluation of two algorithms for a network of coastal HF Radars in the mid-atlantic

bight. Ocean Dynamics, 62(6): 953-968.

Pawlowicz, R., Beardsley, B. and Lentz, S., 2002. Classical tidal harmonic analysis including error estimates in MATLAB using T_TIDE. Computers & Geosciences, 28(8): 929-937.

Roarty, H. et al., 2010. Operation and application of a regional high-frequency radar network in the Mid-Atlantic Bight. Marine Technology Society Journal, 44(6): 133-145.

Roarty, H. et al., 2020. Annual and seasonal surface circulation over the Mid-Atlantic bight continental shelf derived from a decade of high frequency radar observations. Journal of Geophysical Research: Oceans, 125(11): e2020JC016368.

Seroka, G. et al., 2016. Hurricane Irene sensitivity to stratified coastal ocean cooling. Monthly Weather Review, 144(9): 3507-3530.

Seroka, G. et al., 2017. Rapid shelf-wide cooling response of a stratified coastal ocean to hurricanes. Journal of Geophysical Research: Oceans, 122(6): 4845-4867.

Shay, L.K. and Brewster, J.K., 2010. Oceanic heat content variability in the eastern pacific ocean for hurricane intensity forecasting. Monthly Weather Review, 138(6): 2110-2131.

Vincent, E.M. et al., 2012. Assessing the oceanic control on the amplitude of sea surface cooling induced by tropical cyclones. Journal of Geophysical Research: Oceans, 117(C5).

Xu, Y. and Yu, X., 2021. Enhanced atmospheric wave boundary layer model for evaluation of wind stress over waters of finite depth. Progress in Oceanography, 198: 102664.

Zhang, F., Li, M. and Miles, T., 2018. Generation of near-inertial currents on the Mid-Atlantic Bight by hurricane Arthur (2014). Journal of Geophysical Research: Oceans, 123(4): 3100-3116.

Zhang, H. et al., 2016. Upper ocean response to Typhoon Kalmaegi (2014). Journal of Geophysical Research: Oceans, 121(8): 6520-6535.